# Lifetime alcohol use and overall and cause-specific mortality in the European Prospective Investigation into Cancer and nutrition (EPIC) study

Pietro Ferrari,[1] Idlir Licaj,[1] David C Muller,[1] Per Kragh Andersen,[2] Mattias Johansson,[1] Heiner Boeing,[3] Elisabete Weiderpass,[4,5,6,7] Laure Dossus,[8,9,10] Laureen Dartois,[8,9,10] Guy Fagherazzi,[8,9,10] Kathryn E Bradbury,[11] Kay-Tee Khaw,[12] Nick Wareham,[13] Eric J Duell,[14] Aurelio Barricarte,[15,16] Esther Molina-Montes,[17,18] Carmen Navarro Sanchez,[18,19,20] Larraitz Arriola,[16,21] Peter Wallström,[22] Anne Tjønneland,[23] Anja Olsen,[23] Antonia Trichopoulou,[24,25] Vasiliki Benetou,[24] Dimitrios Trichopoulos,[25,26,27] Rosario Tumino,[28] Claudia Agnoli,[29] Carlotta Sacerdote,[30,31] Domenico Palli,[32] Kuanrong Li,[33] Rudolf Kaaks,[33] Petra Peeters,[34] Joline WJ Beulens,[34] Luciana Nunes,[1,35] Marc Gunter,[36] Teresa Norat,[36] Kim Overvad,[37,38] Paul Brennan,[1] Elio Riboli,[36] Isabelle Romieu[1]

▶ Prepublication history and additional material is available. To view please visit the journal (http://dx.doi.org/10.1136/bmjopen-2014-005245).

For numbered affiliations see end of article.

**Correspondence to**
Dr Pietro Ferrari;
ferrarip@iarc.fr

## ABSTRACT

**Objectives:** To investigate the role of factors that modulate the association between alcohol and mortality, and to provide estimates of absolute risk of death.
**Design:** The European Prospective Investigation into Cancer and nutrition (EPIC).
**Setting:** 23 centres in 10 countries.
**Participants:** 380 395 men and women, free of cancer, diabetes, heart attack or stroke at enrolment, followed up for 12.6 years on average.
**Main outcome measures:** 20 453 fatal events, of which 2053 alcohol-related cancers (ARC, including cancers of upper aerodigestive tract, liver, colorectal and female breast), 4187 cardiovascular diseases/coronary heart disease (CVD/CHD), 856 violent deaths and injuries. Lifetime alcohol use was assessed at recruitment.
**Results:** HRs comparing extreme drinkers (≥30 g/day in women and ≥60 g/day in men) to moderate drinkers (0.1–4.9 g/day) were 1.27 (95% CI 1.13 to 1.43) in women and 1.53 (1.39 to 1.68) in men. Strong associations were observed for ARC mortality, in men particularly, and for violent deaths and injuries, in men only. No associations were observed for CVD/CHD mortality among drinkers, whereby HRs were higher in never compared to moderate drinkers. Overall mortality seemed to be more strongly related to beer than wine use, particularly in men. The 10-year risks of overall death for women aged 60 years, drinking more than 30 g/day was 5% and 7%, for never and current smokers, respectively. Corresponding figures in men consuming more than 60 g/day were 11% and 18%, in never and current smokers, respectively. In competing risks analyses, mortality due to CVD/CHD was more pronounced than ARC in men, while CVD/CHD and ARC mortality were of similar magnitude in women.
**Conclusions:** In this large European cohort, alcohol use was positively associated with overall mortality, ARC and violent death and injuries, but marginally to CVD/CHD. Absolute risks of death observed in EPIC suggest that alcohol is an important determinant of total mortality.

## Strengths and limitations of this study

- This study was based on information on dietary and lifestyle exposure collected in a large prospective investigation of European adults.
- Findings are based on 380 395 men and women (among whom 20 453 fatal events occurred) for which information on lifetime alcohol use was available, allowing separate consideration of former drinkers from lifetime abstainers.
- Exclusion of study participants reporting a morbid condition at baseline, and sensitivity analyses excluding the first 3 years of follow-up limited the chance that reverse causality affected the findings.
- Although statistical models included many potentially relevant adjustment factors, residual confounding might partially account for the observed associations.
- Average lifetime alcohol consumption was evaluated in this study, whereas it is possible that specific drinking patterns in particular phases of life, as well as the effect of binge drinking or drinking during meals, may also be of particular relevance for mortality.

## INTRODUCTION

Alcohol intake has been associated with an increased risk of death from a large list of morbid conditions, including digestive tract conditions, liver cirrhosis, chronic pancreatitis, hypertension, injuries and violence.[1–3] In contrast, moderate alcohol drinking was suggested to be associated with a reduction in cardiovascular disease (CVD) mortality.[4 5] As for cancer, the International Agency for Research on Cancer and the World Cancer Research Fund/American Institute for Cancer Research concluded that alcohol use is associated with an increased risk to develop cancers of the upper aerodigestive tract, liver, colorectal and female breast.[6 7] It has been estimated that alcohol accounted for about 2.7 million annual deaths and 3.8% of all deaths worldwide,[8 9] but the impact of alcohol on mortality is differential with respect to specific causes of diseases.[3]

Within the European Prospective Investigation into Cancer and nutrition (EPIC), a recent study showed that heavy alcohol use was associated with a higher risk of death from alcohol-related cancer, external causes and 'other causes', while no associations were observed for coronary heart disease and other cardiovascular diseases.[10]

In this study, we further investigated associations between alcohol use and overall and cause-specific mortality. Notably, potential variability of the relationships with respect to smoking habits, the type of alcoholic beverages and country was explored. The cumulative probabilities of death were estimated for overall mortality and, in a competing risks framework, for specific mortality causes with respect to levels of alcohol, separately in men and women. Furthermore, the burden of alcohol use in relation to a broad group of causes of deaths was evaluated by means of overall estimates of rate advancement periods, with respect to two alternative scenarios.

## METHODS
### Study population

EPIC is an on-going multicentre study that has been described in detail previously.[11] From 1992 to 2000, 521 448 individuals, aged 25–70 years were recruited in the surroundings of 23 centres in 10 European countries. Most of the participants were recruited from the general population residing in a given geographic area, a town or a province. Exceptions were the cohorts of France (female members of a health insurance scheme for school employees), Utrecht (breast cancer screening attendees), Ragusa (blood donors and their spouses) and Oxford (mainly vegetarian and healthy eaters). Some characteristics of the study population in the different participating countries are reported in table 1. Study participants provided informed consent and completed questionnaires on their diet, lifestyle and medical history. The study was approved by the relevant ethical review boards of each centre and the International Agency of Research on Cancer in Lyon, France.[11]

### Dietary and lifestyle assessment

Diet was assessed at enrolment using validated country-specific or centre-specific dietary questionnaires designed to capture habitual consumption over the preceding year. Lifetime alcohol consumption was assessed based on self-reported weekly consumption of wine, beer and liquor at ages 20, 30, 40, 50 years in the lifestyle questionnaire. Information on lifetime alcohol consumption was available for approximately 76% of EPIC participants.[12] Information on smoking status and duration, physical activity during leisure time, prevalent conditions at baseline, educational attainment, anthropometric measures and reproductive history was obtained using lifestyle questionnaires.

### Assessment of causes of death

Vital status and information on cause and date of death were ascertained using record linkage with cancer registries, boards of health and death registries (Denmark, Italy, the Netherlands, Spain, the UK) or by active follow-up (France Germany, Greece). Data were coded using the 10th revision of the International Statistical Classification of Diseases, Injuries and Causes of Death (ICD-10) where the underlying cause is the official cause of death.

In this work, six different causes of deaths were selected: cardiovascular disease (CVD) (I00–I99 excluding I20–I25) and coronary heart disease (CHD) (I20–I25), alcohol-related cancer (ARC), including colorectal cancer (C18–C20), female breast cancer (C50), upper aerodigestive cancers (UADT, including cancer of the mouth (C01–C10 without C08=salivary gland), larynx (C21), pharynx (C11–C14), oesophagus (C15)), violent deaths and injuries (injury, poisoning and certain other consequences of external causes (S00–T98); deaths due to respiratory diseases (J00–J99); a group for all other causes (including external causes of morbidity and mortality (V01–Y98), unknown causes (R96–R99)).

### Statistical analyses

Participants from Denmark (Aarhus, Copenhagen), France, Germany (Heidelberg, Potsdam), Greece, Italy (Florence, Varese, Ragusa, Turin), the Netherlands (Utrecht), Spain (Asturias, Granada, Murcia, Navarra, San Sebastian) and the UK (Cambridge, Oxford) were eligible for this analysis. We excluded the entire cohorts of Naples (Italy), Bilthoven (the Netherlands), Sweden and Norway because no information on past alcohol use was collected (n=118 082). Further exclusions concerned participants with incomplete vital status information (n=928), who had not filled out the dietary or lifestyle questionnaires (n=11 411) and participants whose ratio of energy intake to estimated energy requirement (n=7592), calculated in terms of gender, body weight, height and age, was in the top or bottom 1% in order to partially reduce the impact of outlier values.[13] Participants that at recruitment reported cancer (n=13 283), diabetes (n=11 240), myocardial infarction

**Table 1** Country-specific and sex-specific number of participants (N), person-years (PY), cause-specific and overall number of deaths

| Country | N | PY | CHD* | CVD† | Cancers Breast | UADT‡ | Liver | Colon-rectum | Total§ | Other cancers¶ | Violent and injuries** | Resp†† | Other causes‡‡ | Total |
|---|---|---|---|---|---|---|---|---|---|---|---|---|---|---|
| **Women** | | | | | | | | | | | | | | |
| France | 65 127 | 971 127 | 45 | 202 | 62 | 8 | 4 | 27 | 101 | 678 | 115 | 73 | 1619 | 2833 |
| Italy | 24 956 | 306 244 | 26 | 87 | 71 | 6 | 12 | 53 | 142 | 293 | 30 | 12 | 120 | 710 |
| Spain | 23 616 | 323 027 | 41 | 50 | 51 | 6 | 9 | 47 | 113 | 243 | 46 | 9 | 119 | 621 |
| UK | 50 251 | 651 640 | 387 | 320 | 160 | 30 | 15 | 127 | 332 | 716 | 85 | 161 | 1091 | 3092 |
| The Netherlands | 14 583 | 189 531 | 110 | 137 | 58 | 14 | 7 | 78 | 157 | 370 | 20 | 65 | 209 | 1068 |
| Greece | 14 391 | 143 150 | 139 | 100 | 41 | 2 | 10 | 18 | 71 | 146 | 26 | 27 | 94 | 603 |
| Germany | 27 098 | 307 380 | 56 | 74 | 64 | 12 | 14 | 44 | 134 | 256 | 35 | 25 | 114 | 694 |
| Denmark | 27 773 | 328 375 | 84 | 143 | 128 | 23 | 17 | 110 | 278 | 648 | 51 | 116 | 548 | 1868 |
| All | 247 795 | 3 220 474 | 888 | 1113 | 635 | 101 | 88 | 504 | 1328 | 3335 | 408 | 488 | 3848 | 11 489 |
| **Men** | | | | | | | | | | | | | | |
| France | – | – | – | – | – | – | – | – | – | – | – | – | – | – |
| Italy | 13 471 | 168 992 | 53 | 59 | – | 8 | 10 | 39 | 57 | 240 | 35 | 13 | 131 | 588 |
| Spain | 14 089 | 189 942 | 136 | 88 | – | 34 | 14 | 62 | 110 | 351 | 81 | 42 | 151 | 959 |
| UK | 20 452 | 262 720 | 438 | 229 | – | 41 | 11 | 68 | 120 | 567 | 88 | 171 | 1040 | 2653 |
| The Netherlands | – | – | – | – | – | – | – | – | – | – | – | – | – | – |
| Greece | 9726 | 90 989 | 193 | 141 | – | 12 | 20 | 31 | 63 | 279 | 49 | 54 | 99 | 878 |
| Germany | 19 743 | 221 724 | 167 | 138 | – | 37 | 25 | 65 | 127 | 437 | 84 | 35 | 264 | 1253 |
| Denmark | 24 454 | 282 622 | 273 | 271 | – | 91 | 31 | 126 | 248 | 838 | 111 | 95 | 794 | 2633 |
| All | 101 935 | 1 216 988 | 1260 | 926 | – | 223 | 111 | 391 | 725 | 2712 | 448 | 410 | 2479 | 8964 |

*CHD, coronary heart disease (I20–I25) deaths.
†CVD, cardiovascular disease (I00–I99 except I20–I25) deaths.
‡UADT deaths from upper aerodigestive cancers (including cancer of the mouth (C01–C10 without C08=salivary gland)), larynx (C21), pharynx (C11–C14), oesophagus (C15)).
§Total frequency of alcohol-related cancers.
¶Other cancers: deaths from all other cancers.
**Violent deaths and injuries, including injury, poisoning and certain other consequences of external causes (S00–T98), and external causes of morbidity and mortality (V01–Y98).
††Resp=respiratory diseases (J00–J99).
‡‡All other causes of death.

or heart disease (n=5266) or stroke (n=3246) were excluded from the analyses (n=30 665 in total).

Cox proportional hazard models were used to compute mortality HR, and 95% CIs, for categories of average lifetime alcohol use; never drinkers, 1–4.9 g/day (reference category), 5–14.9, 15–29.9, 30–59.9, ≥60 g/day. In women, the last two alcohol categories were collapsed into a ≥30 g/day group. Time in the study up to death, loss or end of follow-up, whichever came first was the primary time variable. The Breslow method was adopted for handling ties. Models were stratified by centre to control for differences in questionnaire design, follow-up procedures and other centre-specific effects.[13] Systematic adjustments were undertaken for age at recruitment, body mass index and height (continuous), an indicator for participants who quitted alcohol drinking, time since alcohol quitting (continuous), smoking (never, current with 1–15 cigarettes/day, current with more than 15 cigarettes/day, former smoker that quitted less than 10 years before recruitment, former smoker that quitted more than 10 years before recruitment, current smoker of other than cigarettes, unknown (n=8819)), duration of smoking (continuous), age at start smoking (less than 19 years, more than 19 years, unknown (n=39 041)), educational attainment (five categories of level of schooling: none, primary, technical or degree or more, unknown (n=14 223)) as a proxy variable for socioeconomic status, physical activity (inactive, moderately inactive, moderately active, active, unknown (n=328)) and energy intake (continuous). In women the models were further adjusted for menopausal status (dichotomised as natural postmenopausal or surgical vs premenopausal or peri-menopausal, as assessed at baseline), ever use of replacement hormones, and number of full-term pregnancies (nulliparous, one or two children, more than three, unknown (n=6482)). Indicator variables specific to some of the confounding factors were used to model missing values, after checking that the parameters associated with these indicators were not statistically significantly associated with risk of death.

Models for overall and cause-specific mortality were fitted, separately for men and women. An overall test of significance of HRs related to alcohol use was determined by computing p values ($p_{Wald}$) for Wald test statistics compared with a $\chi^2$ distribution with degrees of freedom equal to the number of alcohol categories minus one. The proportional hazards assumption in the Cox model was satisfied and evaluated via inclusion into the disease model of interaction terms between lifetime alcohol and follow-up time. To reduce the chance of reverse causality, sensitivity analyses were run excluding the first 3 years of follow-up. As results were not different from those using the entire cohort, they were not shown. Analyses excluding former drinkers (4% and 5% of the study populations, in men and women, respectively) provided very similar results (results not shown).

### Evaluating heterogeneity

Effect modification in the relation between alcohol and mortality by, in turn, smoking status (never, ever), smoking status (never and current smokers) and recruitment country was assessed. Models with main effects and interaction terms were fitted and compared with models with main effects only. The difference in log-likelihood (likelihood ratio test statistics) was compared to a $\chi^2$ distribution with degrees of freedom equal to the number of interaction terms. HRs for alcohol categories across levels of interacting variables were computed as linear combinations of main effects and interactions. Associations with wine and beer use (each grouped as never, 0.1–2.9 g/day (reference), 3–9.9, 10–19.9, 20–39.9, ≥40 g/day, ≥20 g/day in women) and total mortality were assessed in mutually adjusted models. The difference of association for wine and beer use in relation to overall mortality was assessed by inspecting the significance of the parameter related to the arithmetic difference of wine and beer use (expressed on the log-scale plus 1 to deal with abstainers) in a model that also included their arithmetic sum. When assessing the association for wine (beer) intake, analyses were restricted to moderate lifetime drinkers of beer (wine) and spirits (below 3 g/day).

Flexible parametric survival models[14] with age as the time scale were used to evaluate whether the association between alcohol intake and mortality rate varied by attained age. The origin of the time scale was set to 30 years as the hazard of death is essentially zero prior to that age. The baseline cumulative hazard was modelled using restricted cubic splines with three internal knots placed at evenly spaced centiles of the uncensored log-survival times in order to ensure that an equivalent number of deaths occurred between each knot.[15] Interactions between alcohol intake and the time scale were modelled using restricted cubic splines with one knot placed at the median of uncensored log-survival times. HRs and differences in survival functions were calculated from the fitted models and plotted against attained age, along with CIs calculated based on δ method variance estimates.

Possible departures from linearity in the association between average lifetime alcohol use and total mortality were assessed using fractional polynomials,[16] a subset of generalised linear models in which various powers (−2, −1, −0.5, 0, 0.5, 1, 2, 3) of the covariate(s) of interest are entered into the linear predictor. Fractional polynomials of order two were consistently used in this work for lifetime alcohol use.[17] Non-linearity was tested comparing the difference in log-likelihood of a model with the fractional polynomials with a model with a linear term only to a $\chi^2$ distribution with three degrees of freedom.[16]

### Absolute risks

An extension of the Cox proportional hazards model was employed to fit cause-specific associations between

lifetime alcohol use and cause-specific mortality in a competing risks framework.[18] An augmented data set was created where the initial data set is replicated a number of times equal to the different causes of death. In each replicated data set, competing causes of death were set to censored observations and the analyses were stratified by the event type.[19] The relationship between each confounder variables and cause-specific mortality was assumed homogeneous across causes of death. In this way competing risks were accounted for, and cumulative cause-specific and overall mortality curves were estimated for heavy (greater than 30 and 60 g/day, in women and men, respectively) and moderate (0.1–4.9 g/day) drinkers, separately for never and current smokers.[20] Cumulative mortality curves were obtained for participants aged 60 years, using mean values for continuous confounding factors and average frequencies for categorical confounders.

### Quantifying the alcohol burden

The burden of alcohol on mortality was quantified with estimates of the rate advancement period (RAP),[21] according to two scenarios with threshold levels equal to 5 and 15 g/day. For overall and cause-specific risk of death, RAP were computed, dividing the log(HR) estimate comparing alcohol users above and the threshold with alcohol drinkers between 0.1 g/day and the threshold, by the log of the parameter associated with age. Never alcohol users were not included in the estimation. Associated 95% CIs were also determined. RAP estimates express the impact of a given exposure on the risk of death, by determining the time (in years) by which the risk of death is anticipated for exposed study participants compared with non-exposed.

Statistical tests were two-sided, and p values <0.05 were considered statistically significant. All analyses were performed using SAS V.9.2[22] and Stata V.12.1.[23]

### RESULTS

### Baseline characteristics

The current analysis was based on 247 795 female and 101 935 male study participants. The median age at enrolment was 52 years for women and 53 years for men. Study participants were followed on average 12.6 years, accumulating 4 800 585 person-years, during which a total of 20 453 fatal events were recorded (table 1).

Drinking patterns differed substantially between men and women (table 2). In women, 10% (n=25 146) of participants were lifetime never drinkers, while 45% (n=112 281) and 2% (n=6042) were moderate (0.1–4.9 g/day) and heavy users (>30 g/day), respectively. Conversely, only 1.5% (n=1600) of men reported having never consumed alcohol, 14% (n=14 287) were moderate drinkers, while 29% (n=29 124) were heavy or extreme drinkers (30–59.9 and ≥60 g/day). Furthermore, the vast majority of women who were

regular drinkers (total alcohol intake ≥10 g/day) drank predominantly wine (91%), rather than beer (9%), while regular drinkers in men drank beer (46%) and wine (54%) in similar proportions.

Compared to never and moderate drinkers, women with higher alcohol use had higher levels of education and physical activity, and were more likely to be current smokers or premenopausal/perimenopausal. Never alcohol users were less likely to have used hormonal replacement therapy than alcohol drinkers. In men, the trends were somewhat less apparent. Heavy and extreme alcohol users (≥30 g/day) were more often current smokers, attained lower educational level and had higher energy intake levels, compared with moderate drinkers. Never drinkers were physically less active than alcohol drinkers.

### Lifetime alcohol and total mortality

Lifetime average alcohol use was strongly associated with total mortality, in that never and heavy drinkers (≥30 g/day) had notably higher mortality rates than did light to moderate drinkers (0.1–4.9 g/day), a pattern that was consistently apparent among female and male study participants (figure 1). The HR comparing never and heavy drinkers with moderate drinkers in women was 1.26 (95% CI 1.18 to 1.35) and 1.27 (1.13 to 1.43), respectively. The corresponding HRs among men were 1.29 (1.10 to 1.51) for never drinkers, 1.15 (1.06 to 1.24) for heavy drinkers and 1.53 (1.39 to 1.68) for extreme drinkers (≥60 g/day).

### Lifetime alcohol and cause-specific mortality

In men, extreme alcohol use was associated with mortality due to ARCs (HR$_{\geq60 \text{ vs ref}}$=2.62 1.90 to 3.62), other cancers (1.34 1.13 to 1.59), violent deaths and injuries (1.93 1.27 to 2.91) and other causes (1.98 1.67 to 2.34). With the exception of the category for never drinkers, alcohol intake was not associated with CVD or CHD mortality, in women and men. Among women, heavy drinkers displayed HR$_{\geq30 \text{ vs ref}}$ equal to 1.49 (1.07, 2.06) for ARCs. Respiratory diseases were not associated with lifetime alcohol in women, while results were suggestive of an increased risk in extreme alcohol users compared with moderate users in men (see online supplementary figure S1). Dose–response relationships evaluated with fractional polynomials are displayed in online supplementary figures S2 and S3, for women and men, respectively.

### Evaluating heterogeneity

In both sexes, alcohol-related HRs for overall mortality were of similar magnitude in never and current smokers (table 3). Analyses conducted by smoking intensity (never vs heavy smokers, ie, more than 15 cigarettes/day) produced very similar evidence (results not shown). Cause-specific analyses showed mostly homogeneous alcohol-related HRs by smoking status (results not shown). In women, beer use was more strongly related than wine

**Table 2** Characteristics of the study population at recruitment, according to amount and type of alcohol intake (g/day) in the EPIC study*

| Characteristics | Unit | Never drinkers | Lifetime drinkers 0.1–4.9 | 5–14.9 | 15–29.9 | 30–59.9† | >60† | Total‡ | Wine consumers§ | Beer consumers‡ |
|---|---|---|---|---|---|---|---|---|---|---|
| **Women** | | | | | | | | | | |
| Number of participants | n | 25 146 | 112 281 | 77 147 | 27 179 | 6042 | – | 247 795 | 85 965 | 8748 |
| Person-years | – | 330 854 | 1 460 315 | 998 547 | 352 220 | 78 538 | – | 3 220 474 | 1 124 546 | 110 761 |
| Age at recruitment | Years | 52 (9) | 52 (10) | 51 (10) | 49 (11) | 47 (11) | – | 51 (38–63) | 52 (9) | 46 (12) |
| Lifetime alcohol intake | g/day | 0 (–) | 2 (2) | 9 (3) | 20 (4) | 43 (21) | – | 7 (0–17) | 12 (9) | 11 (9) |
| Educational attainment¶ | % | 14 | 22 | 27 | 33 | 37 | – | 25 | 28 | 28 |
| Current smokers | % | 13 | 14 | 18 | 24 | 31 | – | 17 | 17 | 28 |
| Body mass index | kg/m² | 27 (5) | 25 (5) | 25 (4) | 24 (4) | 24 (4) | – | 25 (20–31) | 24 (4) | 25 (4) |
| Height | cm | 158 (6) | 161 (6) | 162 (6) | 163 (7) | 164 (6) | – | 162 (153–170) | 162 (6) | 163 (7) |
| (Moderately) active | % | 26 | 39 | 44 | 44 | 43 | – | 40 | 42 | 42 |
| Ever use of HRT** | % | 16 | 25 | 29 | 28 | 25 | – | 25 | 50 | 34 |
| Postmenopausal status†† | % | 50 | 49 | 49 | 44 | 38 | – | 48 | 29 | 20 |
| Energy intake | kcal/day | 1848 (537) | 1943 (537) | 2015 (536) | 2090 (552) | 2195 (602) | – | 1978 (542) | 2046 (544) | 1976 (545) |
| **Men** | | | | | | | | | | |
| Number of participants | n | 1600 | 14 287 | 28 875 | 28 049 | 20 788 | 8336 | 101 935 | 26 137 | 22 136 |
| Person-years | – | 19 114 | 171 739 | 345 899 | 333 784 | 247 612 | 98 841 | 1 216 989 | 317 937 | 259 934 |
| Age at recruitment | Years | 53 (11) | 53 (11) | 53 (9) | 52 (9) | 52 (9) | 52 (9) | 53 (41–64) | 53 (9) | 52 (10) |
| Lifetime alcohol intake | g/day | 0 (–) | 2 (2) | 10 (3) | 22 (4) | 42 (8) | 94 (45) | 25 (3–45) | 30 (27) | 22 (25) |
| Educational attainment¶ | % | 21 | 30 | 31 | 32 | 26 | 14 | 29 | 22 | 27 |
| Current smokers | % | 28 | 22 | 25 | 30 | 36 | 49 | 30 | 31 | 33 |
| Body mass index | kg/m² | 27 (4) | 26 (4) | 26 (3) | 27 (3) | 27 (4) | 28 (4) | 27 (22–31) | 27 (4) | 27 (4) |
| Height | cm | 171 (7) | 174 (7) | 175 (7) | 175 (7) | 174 (7) | 172 (7) | 174 (165–183) | 172 (7) | 175 (7) |
| (Moderately) active | % | 42 | 46 | 50 | 52 | 52 | 50 | 50 | 48 | 52 |
| Energy intake | kcal/day | 2284 (675) | 2267 (650) | 2315 (618) | 2417 (622) | 2569 (646) | 2789 (716) | 2427 (656) | 2487 (652) | 2369 (651) |

*Means±SDs are presented for continuous variables, frequencies for categorical variables.
†In women the last alcohol category is ≥30 g/day.
‡For continuous variables (with exception of energy intake), mean (10–90th centile) values are reported.
§Study participants consuming more than 10 g/day of wine (or beer), and consuming less than 3 g/day of beer (or wine).
¶Participants with a university degree or more.
**HRT=hormonal replacement therapy.
††Postmenopausal women plus women who underwent an ovariectomy.

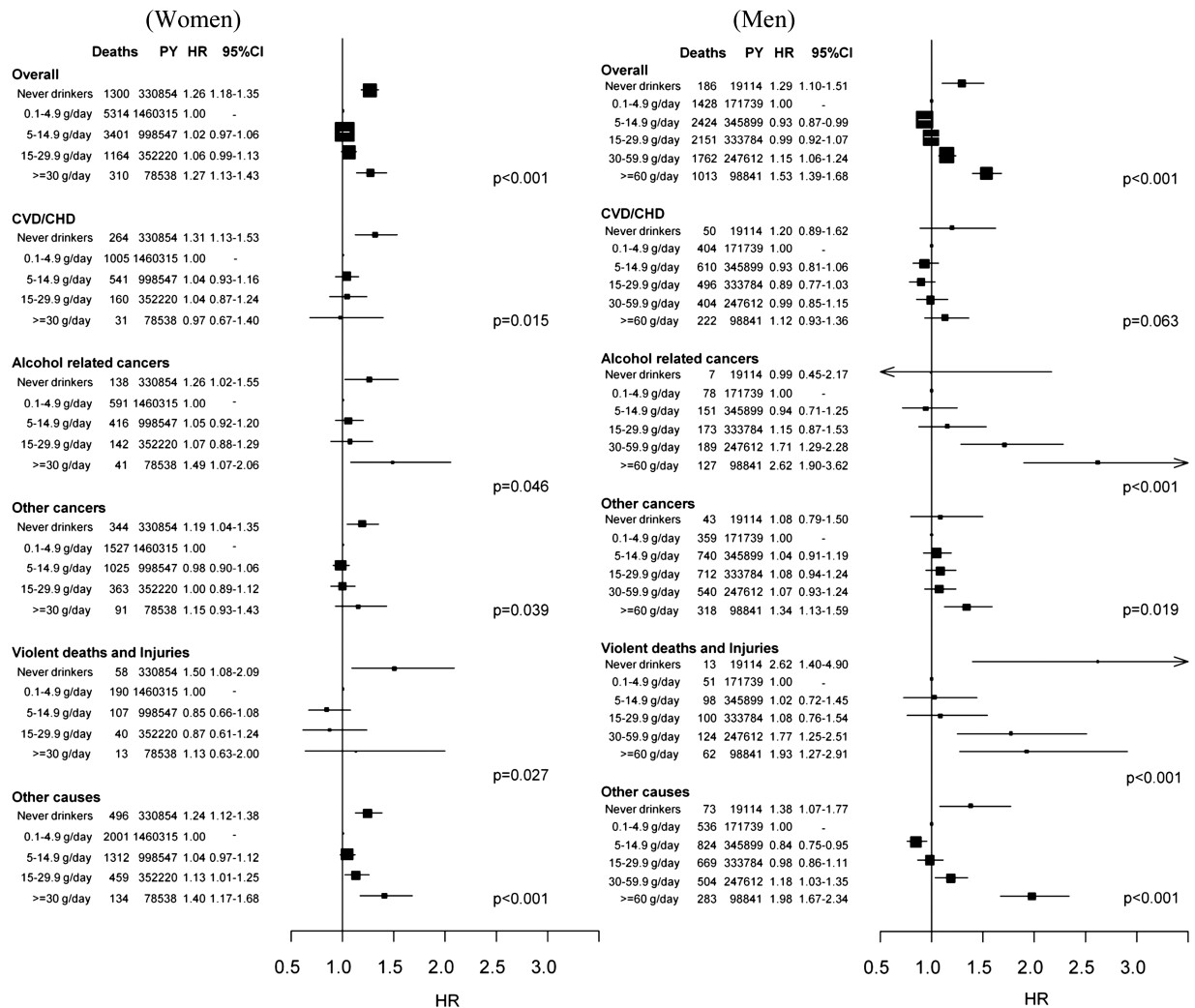

**Figure 1** Number of deaths, person-years (PY) and multivariable HRs (Models were stratified by centre. Systematic adjustment was undertaken for age at recruitment, body mass index and height, former drinking, time since alcohol quitting, smoking status, duration of smoking, age at start smoking, educational attainment and energy intake. In women adjustment was undertaken for menopausal status, ever use of replacement hormones and number of full-term pregnancies.) with 95% CIs and p value of the Wald test for statistical significance for overall and cause-specific mortality by categories of lifetime alcohol use, in women and men.

to overall mortality for amounts greater than 3 g/day compared with the reference category (0.1–2.9 g/day). Lifetime never wine and beer users displayed higher risks than moderate drinkers. The associations between lifetime alcohol and overall risk of mortality were differential across country of origin in men ($p_{heterogeneity}$=0.012) but not in women ($p_{heterogeneity}$=0.511), as reported in online supplementary figures S4 and S5, with stronger relationships observed in Northern European countries compared with Southern European countries.

The $HR_{\geq 60\ vs\ ref}$ for men was more pronounced at earlier ages, and were close to one as attained age approached 90 years ($p_{likelihood-ratio}$ for age-varying vs age invariant parameterisation=0.003); however, extreme male drinkers exhibited lower cumulative survival probability than the reference group throughout the lifespan (see online supplementary figure S6). No such age-varying association was apparent for women ($p_{likelihood-ratio}$=0.80).

### Absolute risks

The 10-year risk of death at the age of 60 years for heavy drinkers was 5% and 7% in women (≥30 g/day), and 11% and 18% in men (≥60 g/day), for never and current smokers, respectively (figure 2). Corresponding figures in moderate drinkers (0.1–4.9 g/day) were 3% and 4% in women, and 5% and 8% in men. Based on a competing risks analysis, it was estimated that, at the age of 60 years, a female lifetime heavy alcohol drinker and smoker had a 10-year risk of death of 1% for ARC, 1.2% for CVD/CHD and 0.2% for violent death and injuries, as displayed in figure 3. Corresponding figures for males (≥60 g/day) were 2.2% (ARC), 5% (CVD/CHD) and 1% (violent death and injuries). Risks for moderate drinkers for ARC, CVD/CHD and violent death and injuries were 0.8%, 1.2%, 0.2% and 0.9%, 4%, 0.3%, in women and men, respectively. Consistently lower risks were observed for never smoker individuals, with

**Table 3** Sex-specific number of deaths, HR* and 95% CI for overall mortality by categories of lifetime alcohol use (g/day), by smoking status (never and current smokers), and type of alcoholic beverage

| | Women | | | | | | Men | | | | | | | |
|---|---|---|---|---|---|---|---|---|---|---|---|---|---|---|
| | Never smokers | | | Current smokers | | | | | Never smokers | | | Current smokers | | |
| Overall | Deaths | HR† | (95% CI) | Deaths | HR† | (95% CI) | p$_{heterog}$‡ | Overall | Deaths | HR† | (95% CI) | Deaths | HR† | (95% CI) | p$_{heterog}$‡ |
| Never | 1009 | 1.34 | (1.24 to 1.45) | 154 | 1.72 | (1.32 to 2.23) | | Never | 84 | 1.50 | (1.19 to 1.21) | 58 | 2.09 | (1.26 to 3.47) | |
| 0.1–4.9 | 3046 | 1 | Ref | 1021 | 1.53 | (1.23 to 1.90) | | 0.1–4.9 | 457 | 1 | Ref | 367 | 1.62 | (1.04 to 2.53) | |
| 5–14.9 | 1550 | 1.04 | (0.98 to 1.11) | 874 | 1.51 | (1.21 to 1.88) | | 5–14.9 | 538 | 0.93 | (0.82 to 1.06) | 799 | 1.45 | (0.93 to 2.25) | |
| 15–29.9 | 397 | 1.04 | (0.94 to 1.16) | 435 | 1.74 | (1.38 to 2.19) | | 15–29.9 | 369 | 1.00 | (0.87 to 1.16) | 927 | 1.66 | (1.06 to 2.58) | |
| ≥ 30 | 82 | 1.29 | (1.03 to 1.61) | 140 | 2.08 | (1.59 to 2.73) | | 30–59.9 | 254 | 1.22 | (1.23 to 1.43) | 857 | 1.83 | (1.17 to 2.84) | |
| p$_{Wald}$§ | | | <0.001 | | | <0.001 | 0.150 | ≥ 60 | 107 | 1.56 | (1.25 to 1.95) | 590 | 2.43 | (1.55 to 3.80) | |
| | | | | | | | | p$_{Wald}$§ | | | <0.001 | | | <0.001 | 0.864 |
| | Wine use | | | Beer use | | | | | Wine use | | | Beer use | | |
| | Deaths | HR¶ | (95% CI) | Deaths | HR¶ | (95% CI) | p$_{difference}$** | | Deaths | HR¶ | (95% CI) | Deaths | HR¶ | (95% CI) | p$_{difference}$** |
| Never | 2156 | 1.15 | (1.09 to 1.22) | 5041 | 1.06 | (1.02 to 1.12) | | Never | 1064 | 1.21 | (1.12 to 1.30) | 975 | 1.07 | (0.98 to 1.16) | |
| 0.1–2.9 | 5109 | 1 | Ref | 5477 | 1 | Ref | | 0.1–2.9 | 3266 | 1 | Ref | 2959 | 1 | Ref | |
| 3–9.9 | 2813 | 0.96 | (0.92 to 1.01) | 787 | 1.15 | (1.07 to 1.24) | | 3–9.9 | 2139 | 0.92 | (0.87 to 0.97) | 2486 | 1.04 | (0.98 to 1.10) | |
| 10–19.9 | 1057 | 1.00 | (0.93 to 1.07) | 147 | 1.50 | (1.27 to 1.77) | | 10–19.9 | 1040 | 0.96 | (0.89 to 1.03) | 1248 | 1.12 | (1.04 to 1.20) | |
| ≥ 20 | 354 | 1.14 | (1.02 to 1.27) | 37 | 1.47 | (1.06 to 2.04) | | 20–39.9 | 814 | 1.03 | (0.95 to 1.13) | 877 | 1.41 | (1.30 to 1.54) | |
| p$_{Wald}$§ | | | <0.001 | | | <0.001 | <0.001 | ≥40 | 641 | 1.22 | (1.10 to 1.35) | 419 | 1.86 | (1.66 to 2.09) | |
| | | | | | | | | p$_{Wald}$§ | | | <0.001 | | | <0.001 | <0.001 |

*Models were stratified by centre. Systematic adjustment was undertaken for age at recruitment, BMI and height, former drinking, time since alcohol quitting, smoking status, duration of smoking, age at start smoking, educational attainment and energy intake. In women adjustment was undertaken for menopausal status, ever use of replacement hormones and number of full-term pregnancies.
†Models included interaction terms between lifetime alcohol use and a smoking indicator (0=never smokers; 1=current smokers), keeping the reference category the group of moderate alcohol users (0.1–4.9 g/day) among never smokers, whereas former smokers and participants with unknown smoking status were excluded.
‡Pheterogeneity: difference in HRs assessed comparing the log-likelihood of models with and without interaction terms between alcohol and smoking status to a four and five degrees of freedom (dof) $\chi^2$ distribution, in women and men, respectively.
§p$_{Wald}$: determined using a Wald test for contrasts according to a $\chi^2$ distribution with four and five degrees of freedom, in women and men, respectively.
¶Models on wine and beer uses were mutually adjusted, and also included spirits/liquors use.
**p$_{difference}$ expresses the difference of associations between wine and beer use, determined evaluating the significance of the parameter estimate $\gamma_2$ in a model that included, other than the list of confounders, the terms $\gamma_1(X_1+X_2)/2+\gamma_2(X_1-X_2)/2$, with $X_1$=log(wine use+1) and $X_2$=log(beer use+1).

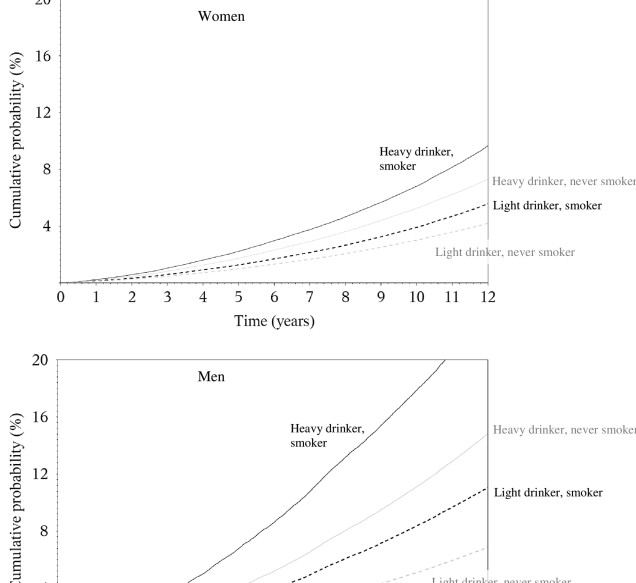

**Figure 2** Sex-specific plots displaying cumulative probabilities of death due to overall mortality, for heavy (=30 g/day in women and=60 g/day in men, continuous line) and moderate lifetime use (0.1–4.9 g/day) (dotted line), in smokers (black line) and never smokers (grey line), for study participants aged 60 years.

estimates equal to 0.5%, 0.7% and 0.1% in women, and 1%, 2.1% and 0.3% in men.

### Rate advancement period

The impact of lifetime alcohol on overall and cause-specific mortality was estimated with RAP values (table 4). In women, RAP for overall mortality were equal to 0.36 years (95% CI −0.05 to 0.76) and 0.83 (0.26 to 1.39), for the 5 and 15 g/day scenario, respectively. In men, RAP values were equal to 0.15 (−0.48 to 0.76) and 1.42 (0.96 to 1.89), for 5 and 15 g/day, respectively. RAP values were sizeable for mortality due to ARC (5.03: 3.07, 7.00) and violent death and injuries (11.83: 3.92, 18.17) in the second scenario.

### DISCUSSION

In this large European prospective study, the association between alcohol use and overall and cause-specific risk of death was evaluated in eight European populations. When accounting for potential confounding factors, average lifetime alcohol use was strongly associated with overall mortality, whereas lifetime never alcohol users consistently displayed a higher risk of death compared with moderate drinkers. These results are in agreement with a recent evaluation of alcohol and cause-specific mortality in EPIC.[10] With respect to this recent study,[10] further analyses were conducted to deeply investigate

the role of factors that modulate the association between alcohol use and the risk of death, notably smoking and types of alcoholic beverage. Estimates of 10-year risk of death in relation to levels of alcohol use were provided.

This study has several strengths. It was conducted using a large prospective cohort, where dietary and lifestyle exposure information were collected on disease-free individuals. Information on lifetime alcohol use was available on 76% of the cohort, allowing separate consideration of former drinkers and lifetime abstainers. Further, exclusion of study participants reporting a morbid condition at baseline, and sensitivity analyses excluding the first 3 years of follow-up suggest that reverse causality is unlikely to have affected the results. One potential weakness of this study is that, although statistical models included many potentially relevant adjustment factors, residual confounding might partially account for the observed associations. In addition, average lifetime alcohol consumption was used throughout this study, whereby it is possible that specific drinking patterns in particular phases of life,[10] as well as the effect of binge drinking or drinking during meals may be of particular relevance for mortality.

A recent Russian study found a strong relationship between vodka and risk of death.[24] While an apparent J-shaped relationship between alcohol use and mortality has been reported,[25 26] the interpretation of this pattern is the subject of some controversy. It has been suggested that alcohol abstinence does not truly entail greater risk of death than moderate use, and that misclassification of alcohol quantity and lack of accuracy in reporting prevalent morbid conditions at baseline in the group of never drinkers[27] could explain the excess risks observed. This reasoning motivated our choice of considering moderate alcohol drinkers, as the reference category throughout this work. Moreover, residual and unmeasured confounding are plausible drivers of the association.[28] These suggestions are supported by our findings that never drinkers are at increased risk of death due to violence and injury. This implausible association casts considerable doubt on the veracity of the apparent increased risk of death among never drinkers.

The overall mortality HR for men with extreme versus moderate alcohol use was greater at younger ages, and approached one as age increased towards 90 years. This result reflects the comparatively low incidence of death through middle age. Consideration of the absolute risk of death, however, suggests that moderate drinkers have a substantial cumulative survival advantage over extreme drinkers throughout the adult lifespan. It has been suggested that wine drinking could be more favourably associated than other alcoholic beverages to the risk of CHD and some cancers.[29–32] In this study beer use displayed more apparent risk patterns than wine consumption, particularly in men. Although we believe that this finding is relevant, we call for cautious interpretations of these results, as the lifestyle profile of wine and beer drinkers is profoundly different. The associations

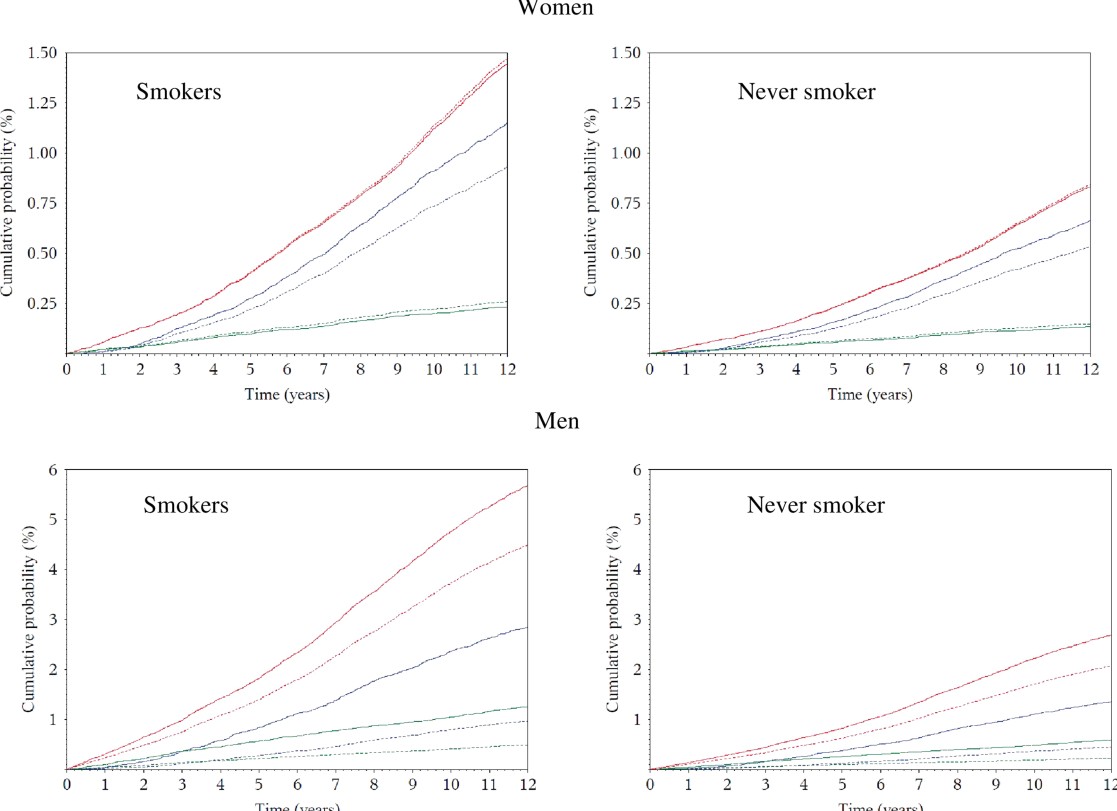

**Figure 3** In competing risks analyses, sex-specific plots displaying cumulative probabilities of death due to CVD/CHD (red), alcohol-related cancers (blue) and violent death and injuries (green), for study participants aged 60 years according to heavy (=30 g/day in women and=60 g/day in men, continuous line) and moderate (0.1–4.9 g/day, dotted lines) lifetime alcohol use in current and never smokers in the EPIC study.

between alcohol and mortality were heterogeneous across countries in men, but not in women. This could be due to the larger amount of alcohol consumed in men than in women, naturally increasing the variability

of exposure and the statistical power to detect associations, and of the larger variability characterising drinking habits in men, such as binge drinking, drinking during meals and other societal aspects. Although no

**Table 4** Sex-specific estimates of rate advancement period (RAP) and associated 95% CI for overall and mortality due to ARCs, CVD/CHD and injuries and violent deaths, related to two scenarios of lifetime alcohol use. RAP estimates express the impact of a given exposure on the risk of death, by determining the time (in years) by which the risk of death is anticipated for study participants exposed, for example, all drinkers more than the threshold (5 or 15 g/day in Scenarios I and II, respectively), compared to non-exposed, that is, individuals drinking between 0.1 g/day and the threshold*

| | Scenario I | | Scenario II | |
| | Threshold 5 g/day | | Threshold 15 g/day | |
| | RAP (years) | 95% CI | RAP (years) | 95% CI |
|---|---|---|---|---|
| Women | | | | |
| Overall | 0.36 | −0.05 to 0.76 | 0.83 | 0.26 to 1.39 |
| CVD/CHD | 0.23 | −0.46 to 0.92 | 0.08 | −0.96 to 1.14 |
| Alcohol-related cancers | 1.28 | −0.86 to 3.41 | 1.90 | −1.00 to 4.81 |
| Injuries and violent deaths | −2.69 | −6.85 to 1.47 | −0.20 | −5.85 to 5.46 |
| Men | | | | |
| Overall | 0.15 | −0.48 to 0.76 | 1.42 | 0.96 to 1.89 |
| CVD/CHD | −0.53 | −1.57 to 0.50 | −0.01 | −0.82 to 0.81 |
| Alcohol-related cancers | 2.59 | −0.30 to 5.49 | 5.03 | 3.07 to 7.00 |
| Injuries and violent deaths | 7.59 | −2.82 to 18.02 | 11.83 | 3.92 to 18.17 |

*Never lifetime alcohol users did not enter into the estimation process.
CVD/CHD, cardiovascular diseases coronary heart disease; RAP, rate advancement period.

heterogeneity was observed by smoking status, mortality among smokers was higher than the mortality among non-smokers, so the absolute increase in risk associated with alcohol intake is more extreme among smokers. This differential increase in cumulative probability of death emphasises the central role of tobacco as a risk factor for mortality, as well as the potential extra harm of increased alcohol consumption.

In the EPIC study, although associations with alcohol were mostly apparent for ARC and violent death and injuries, absolute risks were highest for CVD/CHD in men, while CVD/CHD and ARC risks were of similar magnitude in women. In general, as individuals reporting a prevalent condition at recruitment (either cancer, diabetes, heart attack or stroke) were excluded from the analysis in an effort to minimise reverse causality, our estimates of absolute risks of death are possibly underestimated. RAP values were estimated to appreciate the risk benefit of alcohol drinkers if they were to modify their exposure, according to a counterfactual scenario.[21] [33] Consistently in men and women, RAP values for overall mortality were larger when the reference category was set to 0.1–15 g/day than when using a threshold of 5 g/day, thus indicating that, based on the EPIC study, the benefit for drinkers could be largest if their intake is reduced to levels below 15 g/day. These results are in line with findings of a recent work in the UK population, where the reduction of overall mortality was estimated to be optimised for alcohol reduction up to a median population level of 5 g/day.[34]

In a large prospective study in Europe, lifetime alcohol intake was significantly associated with overall and ARC-specific mortality. In men, positive associations were observed for violent deaths and injuries, while CVD and CHD deaths were not associated with alcohol use among drinkers. Our findings suggest that these associations do not differ between never and current smokers, and were stronger for beer than for wine drinkers.

**Author affiliations**
[1]International Agency for Research on Cancer, Lyon, France
[2]Department of Biostatistics, University of Copenhagen, Copenhagen, Denmark
[3]Potsdam-Rehbrücke Department of Epidemiology, German Institute of Human Nutrition, Nuthetal, Germany
[4]Department of Community Medicine, Faculty of Health Sciences, University of Tromso, The Arctic University of Norway, Tromsø, Norway
[5]Department of Research, Cancer Registry of Norway, Oslo, Norway
[6]Department of Medical Epidemiology and Biostatistics, Karolinska Institutet, Stockholm, Sweden
[7]Samfundet Folkhälsan, Helsinki, Finland
[8]Inserm, Centre for research in Epidemiology and Population Health (CESP), U1018, Nutrition, Hormones and Women's Health team, Villejuif, France
[9]Université Paris Sud, UMRS 1018, Villejuif, France
[10]IGR, Villejuif, France
[11]Cancer Epidemiology Unit, Nuffield Department of Population Health, University of Oxford, Oxford, UK
[12]Department of Public Health and Primary Care, University of Cambridge Addenbrooke's Hospital, Cambridge, UK
[13]Medical Research Council Epidemiology Unit, Institute of Metabolic Science, Addenbrooke's Hospital, Cambridge, UK
[14]Unit of Nutrition, Cancer Epidemiology Research Program, Environment and Cancer, Bellvitge Biomedical Research Institute (IDIBELL), Catalan Institute of Oncology (ICO), Barcelona, Spain
[15]Navarre Public Health Institute, Pamplona, Spain
[16]Consortium for Biomedical Research in Epidemiology and Public Health (CIBER Epidemiología y Salud Pública-CIBERESP), Murcia, Spain
[17]Escuela Andaluza de Salud Pública, Instituto de Investigación Biosanitaria de Granada, Granada, Spain
[18]CIBER Epidemiología y Salud Pública (CIBERESP), Spain
[19]Department of Epidemiology, Murcia Regional Health Council, Murcia, Spain
[20]Department of Health and Social Sciences, Universidad de Murcia, Murcia, Spain
[21]Public Health Division of Gipuzkoa, Instituto BIO-Donostia, Basque Government, Spain
[22]Nutrition Epidemiology Research Group, Department of Clinical Sciences, Lund University, Malmö, Sweden
[23]Danish Cancer Society Research Center, Copenhagen, Denmark
[24]Department of Hygiene, Epidemiology and Medical Statistics, WHO Collaborating Center for Food and Nutrition Policies, University of Athens Medical School, Athens, Greece
[25]Hellenic Health Foundation, Athens, Greece
[26]Department of Epidemiology, Harvard School of Public Health, Boston, Massachusetts, USA
[27]Bureau of Epidemiologic Research, Academy of Athens, Athens, Greece
[28]Cancer Registry and Histopathology Unit, "Civic-MP Arezzo" Hospital, ASP Ragusa, Italy
[29]Epidemiology and Prevention Unit, Fondazione IRCCS Istituto Nazionale dei Tumori, Milan, Italy
[30]Unit of Cancer Epidemiology, AO Citta' della Salute e della Scienza-University of Turin and Center for Cancer Prevention (CPO-Piemonte), Turin, Italy
[31]Human Genetics Foundation (HuGeF), Turin, Italy
[32]Molecular and Nutritional Epidemiology Unit, Cancer Research and Prevention Institute—ISPO, Florence, Italy
[33]Department of Cancer Epidemiology, German Cancer Research Centre, Heidelberg, Germany
[34]Department of Epidemiology, Julius Center for Health Sciences and Primary Care, University Medical Center, Utrecht, The Netherlands
[35]Department of Statistics, Federal University of Rio Grande do Sul, Porto Alegre, Brazil
[36]Department of Epidemiology & Biostatistics, School of Public Health, Imperial College London, London, UK
[37]Department of Public Health, Section for Epidemiology, Aarhus University, Aarhus, Denmark
[38]Department of Cardiology, Aalborg University Hospital, Aalborg, Denmark

**Contributors** PF, IL, IR, PB and HB conceptualised the study and defined the analytical strategy. PF, IL, DCM, PKA and LN performed statistical analyses and provided preliminary interpretation of findings. PF, DCM, MJ, IL and PB contributed by drafting the manuscript. gEW, LD, LAD, GF, KEB, K-TK, EJD, AB, EM-M, CNS, LA, PW, AT, AO, ATR, VB, DT, RT, CA, CS, DP, KLI, RK , PP, JWJB, MG, TN, KO, ER, PB and IR played a key role in the acquisition of the data, and were active in searching for funding to continue the study. With respect to this work, they all critically helped in the interpretation of results, revised the manuscript and provided relevant intellectual input.

**Funding** This work was supported by the Direction Générale de la Santé (French Ministry of Health) (Grant GR-IARC-2003-09-12-01), by the European Commission (Directorate General for Health and Consumer Affairs) and the International Agency for Research on Cancer. The national cohorts are supported by the Danish Cancer Society (Denmark); the Ligue Contre le Cancer, the Institut Gustave Roussy, the Mutuelle Générale de l'Education Nationale and the Institut National de la Santé et de la Recherche Médicale (France); the Deutsche Krebshilfe, the Deutsches Krebsforschungszentrum, and the Federal Ministry of Education and Research (Germany); the Hellenic Health Foundation, the Stavros Niarchos Foundation and the Hellenic Ministry of Health and Social Solidarity (Greece); the Italian Association for Research on Cancer and the National Research Council (Italy); the Dutch Ministry of Public Health, Welfare and Sports, the Netherlands Cancer Registry, LK

Research Funds, Dutch Prevention Funds, the Dutch Zorg Onderzoek Nederland, the World Cancer Research Fund and Statistics Netherlands (the Netherlands); European Research Council-2009-AdG 232 997 and the Nordforsk, Nordic Centre of Excellence programme on Food, Nutrition and Health (Norway); the Health Research Fund, Regional Governments of Andalucýa, Asturias, Basque Country, Murcia (project 6236) and Navarra, Instituto de Salud Carlos III, Redes de Investigacion Cooperativa (RD06/0020) (Spain); the Swedish Cancer Society, the Swedish Scientific Council and the Regional Government of Skåne (Sweden); Cancer Research UK, the Medical Research Council, the Stroke Association, the British Heart Foundation, the Department of Health, the Food Standards Agency, and the Wellcome Trust (UK). The work undertaken by David C Muller was done during the tenure of an IARC, Australia postdoctoral fellowship, supported by the Cancer Council Australia.

**Competing interests**  KEB reports grants from Cancer Research UK, during the conduct of the study. The rest of the authors declared no support from any organisation for the submitted work, no financial relationships with any organisations that might have an interest in the submitted work in the previous 3 years, no other relationships or activities that could appear to have influenced the submitted work.

**Patient consent**  Obtained.

**Ethics approval**  The study was approved by the Ethical Review Board of the International Agency for Research on Cancer, and by the local Ethics Committees in the participating centres.

**Provenance and peer review**  Not commissioned; externally peer reviewed.

**Data sharing statement**  Statistical code is available from the corresponding author by emailing ferrarip@iarc.fr.

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
