## [Reviewer comments · BMJ Open]

Some articles will have been accepted based in part or entirely on reviews undertaken for other BMJ Group journals. These will be reproduced where possible.

ARTICLE DETAILS

TITLE (PROVISIONAL)	Lifetime alcohol use and overall and cause-specific mortality in the European Prospective Investigation into Cancer and nutrition (EPIC) study.
AUTHORS	Ferrari, Pietro; Licaj, Idlir; Muller, David; Kragh-Andersen, Per; Johansson, Mattias; Boeing, Heiner; Weiderpass, Elisabete; Dossus, Laure; Dartois, Laureen; Fagherazzi, Guy; Bradbury, Kathryn; Khaw, Kay-Tee; Wareham, Nicholas; Duell, Eric; Gurrea, Aurelio; MOLINA-MONTES, Esther; Navarro, Carmen; Arriola, Larraitz; Wallström, Peter; Tjønneland, Anne; Olsen, Anja; Trichopoulou, Antonia; Benetou, Vasiliki; Trichopoulos, Dimitrios; Tumino, Rosario; Agnoli, Claudia; Sacerdote, Carlotta; Palli, Domenico; Li, Kuanrong; Kaaks, Rudolf; Peeters, Petra; Beulens, Joline; Nunes, Luciana; Gunter, Marc; Norat, Teresa; Overvad, Kim; Brennan, Paul; Riboli, Elio; Romieu, Isabelle

VERSION 1 - REVIEW

REVIEWER	Rehm, Jürgen CAMH Canada
REVIEW RETURNED	12-Apr-2014

GENERAL COMMENTS	1) The relationship of this paper to Bergmann et al., IJE (2013) should be better clarified. While this paper is cited, it is not clear what exactly were the new objectives of this paper.2) While the authors state as one of the advantages to separate lifetime from former drinkers, these results are not shown. This separation should be the main analyses.3) The limitations acknowledge the potential impact of patterns, which had been modelled in part in Bergmann et al.
--

REVIEWER	Kun Chen Zhejiang University, China
REVIEW RETURNED	14-Apr-2014

GENERAL COMMENTS	In the article entitled "Lifetime alcohol use and overall and cause-specific mortality in the European Prospective Investigation into Cancer and nutrition (EPIC) study", the authors conducted a large European cohort to investigate the role of factors that modulate the association between alcohol and mortality. There are some merits; however, there are also some flaws. 1. In the table 3, for current smokers, which drinking group is the reference group?2. There are some careless grammar mistakes in this paper. The language should be revised.
---

	3. As we know, there are many researches focused on the relationship between alcohol and mortality. So I want to know what's new in this study.
--	---

VERSION 1 – AUTHOR RESPONSE

Reviewer Name J Rehm, CAMH Canada

1) The relationship of this paper to Bergmann et al., IJE (2013) should be better clarified. While this paper is cited, it is not clear what exactly were the new objectives of this paper.

Many new elements characterize this manuscript with respect to the work by Bergmann et al. They are initially listed in the last paragraph of the Introduction section: the deep investigation by smoking status, by alcohol subtype, and by country. In addition, 10-year estimates of cumulative probabilities of death are provided in this work, for three major groups of cause-specific mortality. We believe this is a very good way to exploit the prospective nature of the EPIC data. In line with the Reviewer's point, the innovative elements of this work are now further highlighted in the discussion section (bottom of page 14).

2) While the authors state as one of the advantages to separate lifetime from former drinkers, these results are not shown. This separation should be the main analyses.

We believe that the Reviewer is referring to the following sentence in the Strengths and Limitations section of the Article Summary: "Findings are based on 380,395 men and women (among whom 20,453 fatal events occurred) for which information on lifetime alcohol use was available, allowing separate consideration of former drinkers and lifetime abstainers". Note that we state that the strength is separation of former drinkers from lifetime abstainers, not former drinkers from lifetime drinkers as the Reviewer claims. The sentence was amended to address the reviewer's concern. In general, the analytical strategy we adopted was vastly focused on the dose-response relationship. To do so, we aimed at using all possible available information, including levels of alcohol use of former drinkers. This choice has the merit of providing complementary evidence with respect to the work by Bergmann et al. In any case, when former drinkers were separated out from the main analysis, results were essentially unchanged.

3) The limitations acknowledge the potential impact of patterns, which had been modelled in part in Bergmann et al.

Indeed we agree with the Reviewer's comment, as the paper by Bergman et al. addresses some very important aspects of the relationship between alcohol and mortality with respect of patterns of consumption. A reference to Bergman et al. paper was added in the discussion when listing the potential limitation of our study. Unfortunately it is not possible to address some other important aspects of drinking patterns in EPIC, mainly related to binge drinking or alcohol use during meals.

Reviewer Name Kun Chen, Zhejiang University, China

In the article entitled "Lifetime alcohol use and overall and cause-specific mortality in the European Prospective Investigation into Cancer and nutrition (EPIC) study", the authors conducted a large European cohort to investigate the role of factors that modulate the association between alcohol and mortality. There are some merits; however, there are also some flaws.

1. In the table 3, for current smokers, which drinking group is the reference group?

The reference category in table 3 is the group of moderate alcohol users (0.1-4.9 g/day) among never smokers. A note is now included in the footnote of the table to make it clearer.

2. There are some careless grammar mistakes in this paper. The language should be revised.

The text has been entirely revised to remove grammatical errors.

3. As we know, there are many researches focused on the relationship between alcohol and mortality. So I want to know what's new in this study.

The large size of the study constitutes an important element. In addition, we believe the 10-year

estimates of cumulative probabilities of death for three major groups of cause-specific mortality are innovative. Even more so, considering that such estimates were computed in the context of competing risks analysis. This is a good way to exploit the prospective nature of the EPIC data. We now further highlight these new aspects in the discussion (first paragraph of the discussion).